# M2Edit: Locate and Edit Multi-Granularity Knowledge in Multimodal Large Language Model

## Abstract

Multimodal knowledge editing is an important method for modifying outdated or incorrect knowledge in Multimodal Large Language Models (MLLMs). However, existing datasets for multimodal knowledge editing lack multi-granularity knowledge. In this paper, we present a more realistic dataset called **M2Edit**, which includes three distinct types of knowledge: entity, relation, and action. Additionally, existing knowledge editing methods for MLLMs lack the ability to handle multi-granularity knowledge and generalize to multimodal data. To address these limitations, we propose the multimodal knowledge editing method **MLE**. This approach identifies key knowledge layers within different components and collaboratively edits the various components of MLLMs. As a result, we observe significant improvements in visual generality performance, ranging from 4.8 to 10.8, and achieve the best overall performance on knowledge data of different granularities.

## 1 Introduction

With the continuous development of multimodal large language models (MLLMs) (Li et al. (2023); Alayrac et al. (2022); Zhu et al. (2023); Dai et al. (2023); Liu et al. (2023)), the efficient modification of knowledge within these models, called multimodal knowledge editing (MKE), has garnered widespread attention (Yao et al. (2023)). Studies on MKE (Cheng et al. (2023); Li et al. (2024)) want to directly edit the knowledge within MLLMs, allowing for the addition of new knowledge or the modification of old knowledge. For instance, as illustrated in Figure 1, when an MLLM is asked to describe the content of the image, it might incorrectly interpret the outdated knowledge that "*Obama is the President of the United States*". This outdated knowledge can be updated

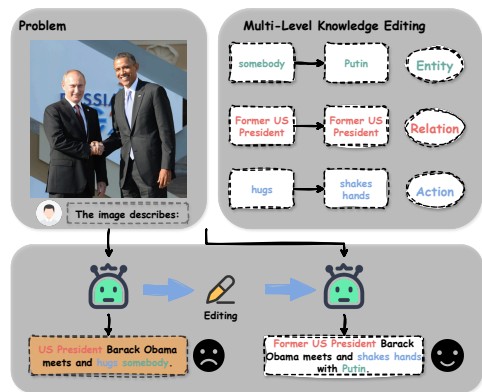

Figure 1: Overview of Multi-Granularity Knowledge Editing. After editing multi-granularity knowledge (i.e., entity, relation, action) in the multimodal large language model, it can solve the problem correctly.

by editing the model. Additionally, if the model does not recognize that the person shaking hands with "*Obama*" is "*Putin*", the new knowledge needs to be injected into the MLLM.

Several research efforts have been dedicated to knowledge editing in MLLMs. There is still a **lack of multi-granular knowledge** in the existing datasets for Multimodal Knowledge Editing (MKE). Specifically, MIKE (Li et al. (2024)) has developed its knowledge editing benchmark based on an entity-level question-answering dataset, which encompasses a significant amount of entity-level

knowledge. However, in real-world scenarios, relying solely on entity-level knowledge proves to be insufficient. As depicted in Figure 1, answering the question correctly, three different types of knowledge (i.e., entity, relation, action) need to be edited. In addition, the effectiveness of various knowledge editing methods cannot be accurately reflected solely by the entity-level knowledge dataset. On the other hand, MMedit (Cheng et al. (2023)) builds its knowledge editing dataset based on open-domain knowledge visual question-answering (Marino et al. (2019)) and image caption datasets (Chen et al. (2015)). They also fail to consider that the knowledge in the dataset should be multi-granular.

To address this challenge, we construct the **M2Edit** (**M**ulti-Granularity **M**ultimodal knowledge **Edit**ing), a dataset contains multi-granularity knowledge. This dataset consists of 3 types of knowledge samples: 35,673 entity samples, 2,167 relation samples, and 4,557 action samples.

However, when applying existing methods (Meng et al. (2022); Mitchell et al. (2022a;b); Cao et al. (2021)) to M2Edit, we encounter two problems: lack of ability to process multi-granularity knowledge and lack of generalization on multimodal data. **Lack of ability to process multi-granularity knowledge:** The existing work has not considered the modeling differences for knowledge of different granularities. However, our experiments have revealed that knowledge of different granularities is stored in distinct regions of MLLMs. Consequently, the existing methods for modeling knowledge are imprecise and lack precision. **Lack of generalization on multimodal data:** While existing methods have shown some effectiveness when directly transferring editing methods from the text modality to existing datasets, they exhibit insufficient generalization on multimodal data. MLLMs are more complex than LLMs (Yao et al. (2023)), as they typically comprise multiple components, including an LLM, a visual encoder, and a multimodal interface. Failing to edit these modules simultaneously is likely to result in poor performance on multimodal data, as confirmed by our experiments.

To overcome the above two challenges, we propose a novel knowledge editing method named **MLE** (**M**ultimodal **L**ocation-based **E**diting). To handle the problem of Lack of ability to process multi-granularity knowledge, MLE sequentially identifies key knowledge layers within the three components of MLLMs for different types of knowledge. To overcome the challenge of lack of generalization on multimodal data. Subsequently, MLE collaboratively edits these key knowledge layers in the three components by the least squares-based method to obtain better generality on multimodal data. Our contributions can be summarized as follows:

- To the best of our knowledge, we are pioneers in advocating for a differentiated treatment of various types of knowledge within MLLMs during knowledge editing. To substantiate this, we have developed a **M**ulti-Granularity **M**ultimodal knowledge **Edit**ing dataset (**M2Edit**), which incorporates three types of knowledge.

- We design a novel multimodal knowledge locate then edit method (**MLE**), which can locate different knowledge in MLLMs to better process multi-granularity data and collaboratively edit different components of MLLMs to achieve superior generalization.

- The experimental results demonstrate the effectiveness of our proposed method compared to Baselines. Additionally, these results validate the differences in the storage of different types of knowledge within the components of MLLMs. The code will be provided as an attachment.

## 2 METHODOLOGY

### 2.1 TASK DEFINITION

For a multimodal large language model (MLLM) (Cui et al. (2024)), let $\Theta$ denote it. An MLLM ($\Theta$) often contains three components: a visual encoder for encoding images, a multimodal interface for converting visual information into a large language model (LLM) space, and an LLM for processing information from images and text simultaneously. Let $\Theta = \{\theta_{ve}, \theta_{mi}, \theta_{llm}\}$ be the components parameters. For a multimodal knowledge editing dataset $\mathcal{D} = \{(x_i, v_i, y_i) | i \in [1, N]\}$, where $x_i, v_i, y_i$ represent the input text prompt, image and editing target respectively, and $N$ represents the number of samples in the dataset. For one sample $(x_i, v_i, y_i)$, the after editing MLLM denotes to $\hat{\Theta}$. The goal of knowledge editing (Yao et al. (2023)) is to successfully output the editing target after

editing (**Reliability**) and to have universality on similar samples (**Generality**) and should have no effect on irrelevant samples (**Locality**).

**Reliability.** Editing reliability needs model to answer the knowledge problem to $y_i$. Specifically, to evaluate the reliability $\mathbf{O}^{rel}(\hat{\Theta})$ of the editing methods can be expressed by the following formula:

$$\mathbf{O}^{rel}(\hat{\Theta}) = \mathbb{E}_{(x_i,v_i,y_i)\in\mathcal{D}}[\mathbf{I}(\hat{\Theta}(x_i,v_i)=y_i)], \tag{1}$$

where $\mathbf{I}(\cdot)$ denotes the indicator function.

**Generality.** Editing generality needs model to answer similar questions about the same knowledge to $y_i$. Following MMEdit (Cheng et al. (2023)), the generality of the editing method is tested from two perspectives: Visual generality ($\mathbf{O}_v^{gen}(\hat{\Theta})$): samples similar to the original image (i.e., $(x_i, v_j, y_i)$ $s.t.$ $v_j \sim v_i$), which can be calculated as

$$\mathbf{O}_v^{gen}(\hat{\Theta}) = \mathbb{E}_{(x_i,v_i,y_i)\in\mathcal{D}}[\mathbf{I}(\hat{\Theta}(x_i,v_j)=y_i)]. \tag{2}$$

Text generality ($\mathbf{O}_t^{gen}(\hat{\Theta})$): samples similar to the original prompt (i.e., $(x_j, v_i, y_i)$ $s.t.$ $x_j \sim x_i$), which can be calculated as

$$\mathbf{O}_t^{gen}(\hat{\Theta}) = \mathbb{E}_{(x_i,v_i,y_i)\in\mathcal{D}}[\mathbf{I}(\hat{\Theta}(x_j,v_i)=y_i)]. \tag{3}$$

**Locality.** The locality of editing methods is evaluated by the MLLM can maintain its original output on irrelevant samples, which can be calculated as follows:

$$\mathbf{O}^{loc}(\hat{\Theta}) = \mathbb{E}_{(x_k,v_k)\in\mathcal{D}}[\mathbf{I}(\hat{\Theta}(x_k,v_k)=\Theta(x_k,v_k))]$$
$$s.t.\ (x_k,v_k) \perp (x_i,v_i), \tag{4}$$

where $\perp$ denotes the two samples are unrelated.

## 2.2 M2EDIT DATASET

In order to overcome the challenge of existing multimodal knowledge editing datasets' lack of multi-granularity knowledge, we construct the M2Edit dataset, which consists of three types of knowledge samples: entity, relation, and action. The overall statistics of the M2Edit dataset are shown in Table 1.

**Entity data.** M2Edit entity data is built by filtering samples from the Oven dataset (Hu et al. (2023)), where each image is linked to a Wikipedia entity via a text query. We select "(image, question, answer)" triples with single-word entity names and manually choose ques-

| Knowledge Type | Entity | Relation | Action |
|---|---|---|---|
| **#Entities** | 877 | 1,403 | 2,850 |
| **#Relations** | - | 6 | - |
| **#Actions** | - | - | 47 |
| **#Images** | 89,182 | 6,017 | 4,557 |
| **#Questions** | 179 | 30 | 235 |
| **#Samples** | 35,673 | 2,167 | 4,557 |

Table 1: Statistics of **M2Edit** dataset. M2Edit contains instances involving three types of knowledge: entity, relation, and action.

tions with at least 5 synonymous queries and entities with over 5 related images for the generality evaluation. As shown in Figure 2 top part, each question contains one entity knowledge, and we replace the edit target with a similar word to ensure models do not contain this knowledge in advance. As illustrated in Figure 2 top part, each question only contains one entity knowledge. For example, the entity "*capybara*" has some related images and can be answered through some synonym questions. Besides, to ensure that all models do not contain this knowledge in advance, we replace the edit target with a similar word. For instance, "*koala*" and "*capybara*" belong to the same category "*animal*", so this example adopts "*koala*" as the editing target. And adopts different categories of entity problems to evaluate the locality.

**Relation data.** M2Edit relation data is built from the FB15k-237-IMG dataset (Liu et al. (2019); Bordes et al. (2013)), a subset of Freebase (Bollacker et al. (2008)), which automatically assigns images to entities from the Internet. We filter triples with simple and unambiguous tail entities and select triples with at least 3 images related to the head entity for visual generality evaluation. To construct text generality sample sets, we use ChatGPT to generate and paraphrase relation questions.

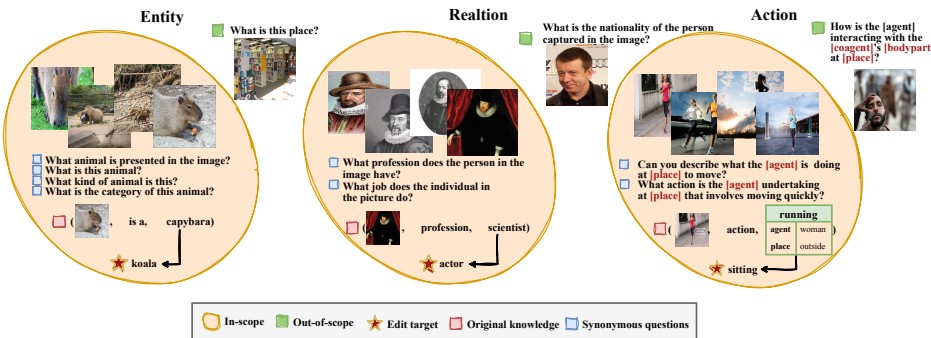

Figure 2: Editing examples for the three knowledge types of **M2Edit**. After editing the MLLMs, the in-scope samples need to be generalizable, and the out-of-scope samples should not be unchanged. For action samples, the semantic slots are filled with specific objects in the image.

As illustrated in Figure 2 middle part, each problem contains knowledge about one relation and two entities. The head entity "*Francis Bacon*" can be represented by multiple images, and the relation "*Profession*" can be represented by some synonym questions. Similarly, we also replace the tail entity with another similar entity to ensure that the knowledge model is free. And adopts different relation problems to evaluate the locality.

**Action data.** M2Edit relation data is based on the ImSitu (Yatskar et al. (2016)) dataset, where each image often depicts a primary action, and provides annotations for the entities involved in the action. We manually select action verbs with clear definitions and use ChatGPT to connect roles in the action schema to form questions and paraphrase them for text generality evaluation. To construct the visual generality set, we select multiple synonymous images from the dataset. As illustrated in Figure 2 bottom part, each problem contains knowledge about one action and a lot of entities involved. The red words represent the semantic slots in the question, which for each image will be filled by the specific entities involved. For example, the "[agent]" of the "*running*" that happened in the image is "*a woman*". Similarly, we also replace the action verb with another verb to be the edit target. And adopts different verb problems to evaluate the locality.

We divide the data into training and testing sets at a 4:1 ratio to accommodate methods that require training.

### 2.3 CASUAL TRACING FOR MULTIMODAL LARGE LANGUAGE MODEL

We apply Causal Mediation Analysis (Shanmugam (2001); Vig et al. (2020)) to track the causal impact of the internal components of the MLLMs, which plays a role in producing answers with multi-granularity knowledge. To trace the important state of the model always needs to take three runs: a clean run that the model can answer the question correctly with normal input, a corrupted run that corrupts the input to make the model get corrupted output, a corrupted-with-restoration run that restores a certain state to judge the restoring of the output. After corrupted-with-restoration run, if the probability of producing the correct answer increases (indirect effect), then the causal relationship between this state and the final result is considered strong. Otherwise, it is considered weak. For detailed procedures, please refer to Appendix A.

### 2.4 MULTIMODAL LOCATE THEN EDIT METHOD

To address the limitation of existing knowledge editing methods that cannot handle multi-granularity knowledge and lack of generalization on multimodal data, we propose a method called MLE (Multimodal Location-based Editing). MLE focuses on different components of the MLLMs, first identifying the specific locations of different knowledge within the model (key knowledge layer), and then performing the least squares-based method to edit them collaboratively. The overall architecture of the model is shown in Figure 3.

Figure 3: The overall architecture of **MLE**. The MLE multimodal knowledge editing framework locates the key knowledge layers storing knowledge in different components of the MLLMs through similar knowledge, then edits the key knowledge layers through least squares fitting expected output ($z$), and finally evaluates the editing results based on four editing evaluation indicators.

### 2.4.1 LOCATE KEY KNOWLEDGE LAYERS

For a knowledge editing sample $s_i = (x_i, v_i, y_i)$, the key layers for storing knowledge (Key Knowledge Layer) in different components are located in turn. First, we will use the MLLM to represent the samples in a specific training set, which can be $\mathcal{M}(x_i, v_i)$. Then, we will apply K-means clustering to these representations to create $k$ clustering center samples as Knowledge Centers $C = \{c_j = (x_j, v_j, y_j) | j \in [1, k]\}$. In addition, we define **Edit Score** to be used to measure the success of editing, which can be

$$\textbf{Edit Score} = \frac{4}{\frac{1}{\mathbf{O}^{rel}} + \frac{1}{\mathbf{O}^{gen}_v} + \frac{1}{\mathbf{O}^{gen}_t} + \frac{1}{\mathbf{O}^{loc}}}. \tag{5}$$

After that, MIE edits each knowledge center sample in each layer from each component of MLLM. The editing layer combination with the maximum Edit Score, that is, the Key Knowledge Layer, is calculated as the most effective editing way for this cluster. The above process can be expressed as

$$L_{key}(c_j) = (r_j, s_j, t_j) = \max_{r,s,t}(\textbf{Edit score}(\hat{\Theta}_{r,s,t}(c_j)))$$
$$r \in [1, L_{llm}], s \in [1, L_{ve}], t \in [1, L_{mi}] \tag{6}$$

where $r_j, s_j, t_j$ represents for a center knowledge sample $c_j$ only editing the $r_j$-th layer of LLM, $s_j$-th layer of the visual encoder, and $t_j$-th layer of the multimodal interface can get the highest Edit Score. Afterward, for a sample in the test set $a_i$, we calculate its cosine similarity with the samples in the knowledge center set to find the closest sample. We then use the Key Knowledge Layer of that center sample for knowledge editing, which can be formulated as

$$L_{key}(a_i) = L_{key}(c_j)$$
$$j = \max_j \frac{\mathcal{M}(a_i)\mathcal{M}(c_i)}{|\mathcal{M}(a_i)||\mathcal{M}(c_i)|}, \tag{7}$$

where $\mathcal{M}(\cdot)$ denotes the representation from MLLM of the sample $a_i$.

### 2.4.2 EDIT KEY KNOWLEDGE LAYER

After identifying the key layers, inspired by A, we can use the least squares-based method for model knowledge editing. We sequentially edit the model using the order of the $r$-th layer of LLM, the $s$-th layer of the visual encoder, and the $t$-th layer of the multimodal interface. Specifically, given some

pairs $(a_i, b_i)$ expressing the same knowledge, where $a_i = (x_i, v_i)$ is the input sample, $b_i$ is the edit target, for the parameter matrix $W$, to update the parameter, it should solve the optimization problem:

$$\min_W \sum_{i=1}^{N} ||Wk_i - z_i||_2^2 + \lambda ||W - W'||_2^2, \tag{8}$$

where $\lambda$ is a regularizer, and $W'$ is original parameter, $k_i$ is the input vector of this layer corresponding to $a_i$ and $z_i$ is the expected output vector corresponding to $b_i$, $N$ is the number of pairs. The optimization problem has a closed-form solution, which can be expressed as the following:

$$W = (\lambda W' + \sum_{i=1}^{N} z_i k_i^T)(\lambda I + \sum_{i=1}^{N} k_i k_i^T)^{-1}, \tag{9}$$

where $I$ denotes the Identity Matrix.

---

**Algorithm 1** Multimodal Locate Then Edit Algorithm

---

**Require:** Training Samples $\mathcal{D}^{\mathcal{T}} = \{(x_i, v_i, y_i) | i \in [1, N]\}$, Testing Samples $\mathcal{D}^{\mathcal{I}} = \{(x_i, v_i, y_i) | i \in [1, M]\}$, Center Number $k$

    ***For Training Samples***
1: Apply K-means clustering to $\mathcal{D}^{\mathcal{T}}$ to get **Knowledge Center** $C = \{c_j = (x_j, v_j, y_j) | j \in [1, k]\}$
2: Initialize the **Key Knowledge Layer** set $L_{key}$
3: **for** $c_j$ in $C$ **do**
4:     **for** $r$ in $[1, L_{llm}]$ and $s$ in $[1, L_{ve}]$ and $t$ in $[1, L_{mi}]$ **do**
5:         Edit the $r$-th layer of LLM, $s$-th layer of vision encoder and $t$-th layer of multimodal interface of MLLM to obtain $\hat{\Theta}_{r,s,t}(c_j)$    *# According to Equation 9*
6:         Calculate the editing score of this combination
7:     **end for**
8:     Calculate the combination of layers $(r_j, s_j, t_j)$ that can maximize the editing score for knowledge $c_j$
9:     Add $(r_j, s_j, t_j)$ to $L_{key}$    *# According to Equation 6*
10: **end for**
    ***For Testing Samples***
11: **for** $a_i$ in $\mathcal{D}^{\mathcal{I}}$ **do**
12:     Calculate the most similar $c_j$ in $C$    *# According to Equation 7*
13:     $L_{key}(a_i) = L_{key}(c_j) = (r_j, s_j, t_j)$
14:     Edit MLLM to obtain $\hat{\Theta}_{r,s,t}(a_i)$    *# According to Equation 9*
15: **end for**
**Ensure:** New Demo Bank $D$

---

The overall process of the proposed method **MLE** is shown in Algorithm 1.

## 3 EXPERIMENTS

### 3.1 IMPLEMENTATION DETAILS

The editing MLLMs in the experiment are BLIP2-OPT 6.7B and MiniGPT4. **BLIP2-OPT** (Li et al. (2023)) adopts a frozen visual transformer (VIT) in EVA-CLIP, frozen OPT as the LLM, and trains a Query Transformer (Q-Former) to connect visual representation with language representation. **MiniGPT4** (Zhu et al. (2023)) is similar to BLIP2, utilizing the same frozen VIT in EVA-CLIP, the same Q-Former and addition linear layer as the multimodal interface, and a frozen Vicuna (Touvron et al. (2023)) as the LLM.

To simplify the calculation process and according to the key-value theory (Geva et al. (2021)), we only consider modifying the parameter of the linear mapping matrix $W$ for the output of each transformer layer. The hyperparameter knowledge centers $k$ is set to 50. We adopt BLIP2-FlanT5xxl as the MLLM to calculate the similarity between samples. In addition, we randomly choose one similar image sample for visual generality evaluation and one synonymous prompt for text generality evaluation. ALL experiments are conducted using NVIDIA GeForce RTX 3090 GPUs.

| Method | Entity | | | | Relation | | | | Action | | | |
|---|---|---|---|---|---|---|---|---|---|---|---|---|
| | R | T-G | V-G | L | R | T-G | V-G | L | R | T-G | V-G | L |
| **BLIP2-OPT** | | | | | | | | | | | | |
| **FT** | 70.2 | 30.5 | 20.3 | 46.9 | 54.3 | 23.8 | 12.4 | 55.9 | 80.6 | 42.4 | 12.4 | 60.4 |
| **KE** | 74.1 | 70.0 | 60.8 | 88.4 | 65.8 | 59.1 | 43.6 | 90.2 | 85.4 | 84.4 | 45.2 | 86.5 |
| **MEND** | 90.7 | 85.0 | 67.4 | 89.6 | 80.4 | 77.4 | 55.3 | 95.3 | 98.2 | 96.5 | 51.4 | 94.3 |
| **SERAC** | 89.2 | 88.7 | 60.1 | 90.6 | 75.6 | 70.3 | 42.3 | **96.2** | 99.0 | 95.3 | 55.2 | 95.6 |
| **ROME** | 80.4 | 73.4 | 58.8 | **91.2** | 69.2 | 63.7 | 32.5 | 94.2 | 93.7 | 90.2 | 52.5 | 93.2 |
| **MLE** | **93.2** | **91.7** | **76.2** | 90.8 | **88.4** | **82.0** | **64.1** | 94.3 | **99.2** | **98.4** | **60.4** | **96.1** |
| **MiniGPT4** | | | | | | | | | | | | |
| **FT** | 22.2 | 10.2 | 5.6 | 40.6 | 17.7 | 14.7 | 1.2 | 53.2 | 26.1 | 21.9 | 3.7 | 70.5 |
| **KE** | 76.7 | 69.5 | 60.6 | 87.6 | 66.8 | 56.4 | 42.3 | 88.1 | 86.0 | 82.9 | 44.3 | 84.9 |
| **MEND** | 92.2 | 83.5 | 68.8 | 90.6 | 80.2 | 79.1 | 55.7 | **98.2** | 98.3 | 98.7 | 52.1 | 96.4 |
| **SERAC** | 91.5 | 88.4 | 60.5 | 90.5 | 79.5 | 72.7 | 45.2 | 97.9 | **99.5** | 97.7 | 57.6 | 94.9 |
| **ROME** | 81.9 | 74.7 | 61.4 | 91.1 | 70.9 | 66.2 | 32.3 | 94.8 | 95.7 | 90.9 | 34.0 | 95.4 |
| **MLE** | **92.9** | **91.8** | **78.6** | **92.6** | **91.4** | **81.7** | **66.5** | 96.3 | 99.4 | **99.0** | **62.0** | **97.9** |

Table 2: Main Multimodal Knowledge Editing Result on the **M2Edit** dataset. R refers to reliability, T-G refers to text generality, V-G refers to visual generality, and L refers to Locality. The upper part shows the results on BLIP2-OPT (Li et al. (2023)) and the lower part on MiniGPT4 (Zhu et al. (2023)).

## 3.2 BASELINES

We evaluate the knowledge editing methods implemented in the EasyEdit (Wang et al. (2023)) toolkit as baselines.

**FineTune (FT)**. It directly fine-tunes all parameters of the last layer of the model for editing samples.

**Model Editor Networks with Gradient Decomposition (MEND)** (Mitchell et al. (2022a)). It learns to efficiently locate knowledge in the LLM, and the knowledge is edited by leveraging the low-rank decomposition of gradients.

**Semi-Parametric Editing with a Retrieval-Augmented Counterfactual (SERAC)** (Mitchell et al. (2022b)). It is a memory-based editing method, which consists of a scope classifier, a base model, and a counterfactual model. For a new sample, the scope classifier is used to determine whether it is in the memory cache, and then the sample in the cache that is most similar to the sample is input into the counterfactual model to obtain the result.

**Knowledge Editor (KE)** (Cao et al. (2021)). It locates the knowledge via a hypernetwork (a bidirectional-LSTM) and predicts parameter updates at inference time via constrained optimization.

**Rank-One Model Editing (ROME)** (Meng et al. (2022)). It locates the knowledge in LLM via Causal Mediation Analysis, the sixth layer of MLP of LLM is updated by the least squares-based method.

## 3.3 COMPARISONS EDITING METHODS

Table 2 shows that our method (MIE) outperforms other methods on all knowledge types of data of M2Edit in most indicators, which demonstrates the effectiveness of our approach. In addition, from the table, we notice:

- Our method achieves effective knowledge editing performance across a wide range of metrics and different types of knowledge data. This indicates that our method can dynamically adapt to different types of knowledge data and effectively edit all three components simultaneously.

- Our method shows the highest improvement in visual generality compared to the baseline model (with improvements ranging from 4.4 to 10.8 in different settings). This demonstrates that collaborative editing of different components of the MLLM can effectively enhance the model's ability to generalize images, addressing the issue of insufficient generalization in the editing.

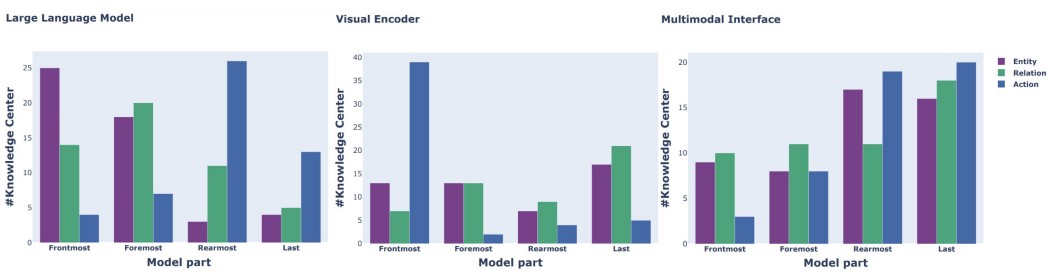

Figure 4: Causal Tracing Results for the LLM MLP of MLLM. The horizontal axis represents different layers, while the vertical axis represents the input characters. The intensity of the bars indicates the probability of generating the correct answer (after causal intervention). Knowledge of different granularities (i.e., entity, relation, action) is scattered in different layers in the LLM.

Figure 5: The distribution of the layers that need to be edited for the knowledge centers in the four parts of MLLM components.

## 3.4 DISTRIBUTION OF KNOWLEDGE IN MLLMS

We conduct the Causal Mediation Analysis on different components of the BLIP2-OPT and found that the storage of different knowledge varies across these components. Particularly in the LLM, different knowledge is stored hierarchically. As shown in Figure 4, it illustrates the AIE (average indirect effect) of the state in the MLP (Multilayer Perceptron) of LLM under different knowledge types. Entity-related knowledge tends to be stored in the foremost part of the LLM, while relation-related knowledge is stored in the foremost section, and event-related knowledge is stored in the rearmost part of the large model.

This conclusion is further supported by the selection of key knowledge layers. We divide the layers in different components of MLLM (BLIP2-OPT 6.7B) into four parts (Frontmost, Foremost, Rearmost, and Last). As shown in Figure 5, it illustrates the selection of different layers in various components of the MLLM as key knowledge layers for different knowledge center samples. It can be observed that in LLM, entity knowledge samples tend to select layers in the Frontmost part, relation knowledge samples tend to select layers in the Foremost part, and action knowledge samples tend to select layers in the Rearmost part. And in the other two components, the editing layers of different knowledge are also different.

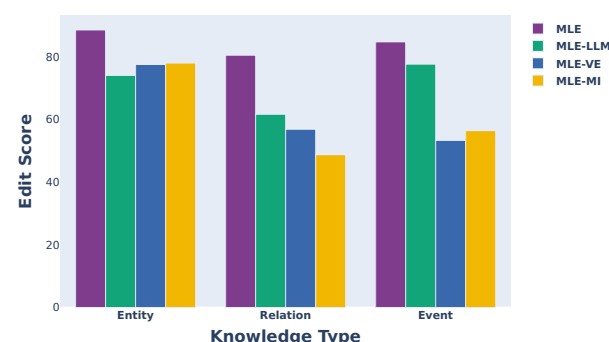

Figure 6: The result of MLE edits different components of BLIP2-OPT 6.7B.

| Method | Entity | | | | Relation | | | | Action | | | |
|---|---|---|---|---|---|---|---|---|---|---|---|---|
| | R | T-G | V-G | L | R | T-G | V-G | L | R | T-G | V-G | L |
| **BLIP2-OPT** | | | | | | | | | | | | |
| FT | **67.4** | 20.2 | 15.6 | 26.4 | 53.2 | 18.7 | 8.5 | 40.2 | **81.3** | 32.6 | 8.9 | 43.3 |
| MEND | 48.1 | 44.2 | 32.5 | 80.4 | 42.0 | **38.6** | 31.8 | **83.1** | 73.2 | 65.3 | 35.4 | 90.4 |
| ROME | 45.4 | 41.8 | 26.9 | 82.5 | 38.3 | 35.3 | 35.0 | 79.5 | 76.7 | 63.2 | 41.2 | **91.2** |
| MLE | 65.9 | **45.2** | **46.3** | **83.1** | **47.2** | 37.2 | **43.3** | 80.5 | 77.3 | **66.8** | **54.8** | 91.2 |
| **MiniGPT4** | | | | | | | | | | | | |
| FT | 24.2 | 5.8 | 5.2 | 26.3 | 15.0 | 4.7 | 1.4 | 38.2 | 28.9 | 22.3 | 5.4 | 54.3 |
| MEND | 53.7 | 50.2 | 34.4 | 82.4 | 46.7 | 38.4 | 24.7 | 88.2 | 63.4 | 55.3 | 43.2 | 92.3 |
| ROME | 55.2 | 48.6 | 32.4 | **84.0** | 48.2 | 39.1 | 27.2 | 89.2 | 72.3 | 59.4 | 48.9 | 93.3 |
| MLE | **61.3** | **51.9** | **43.8** | 82.6 | **51.3** | **39.5** | **34.3** | **90.1** | **74.7** | **61.5** | **53.7** | **93.5** |

Table 3: Batch Editing Results in **M2Edit** for Multimodal Knowledge Editing (The editing of 500 samples in a single batch).

## 3.5 THE IMPORTANCE FOR EDITING DIFFERENT COMPONENTS

As shown in Figure 6, it demonstrates the impact of editing a single component on the editing of three types of knowledge. We found that editing the LLM yields better performance than other components for all types of knowledge, which may indicate that the large model stores a significant amount of knowledge. For entity-related knowledge, the decrease in performance is relatively minimal when editing other components, while for action-related knowledge, the decrease is the most significant. This suggests that a majority of action-related knowledge is stored in the LLM, while entity knowledge is stored relatively scattered.

## 3.6 BATCH EDITING RESULTS

Following the batch editing approach(Meng et al. (2023)), we evaluated the performance of our method after modifying 500 samples, as shown in Figure 3. The results demonstrate that our method still achieves overall performance superior to the baseline, particularly in terms of visual generality performance. However, since our approach is not specifically designed for batch editing, its performance does experience some decline. Nonetheless, we consider this level of degradation to be within an acceptable range.

## 4 RELATED WORK

### 4.1 MODEL KNOWLEDGE EDITING

Both the number of parameters and the amount of training data used in large language models (LLMs) are increasing (Sevilla et al. (2022)). Knowledge is constantly evolving, and for new knowledge that is not present in the model, some researchers are interested in studying knowledge editing (Meng et al. (2022; 2023); Mitchell et al. (2022a)) techniques that involve precisely incorporating knowledge entries into the model without affecting its original performance. ROME (Meng et al. (2022)) and Memit (Meng et al. (2023)) try to locate the knowledge in LLM and then edit them. KE (Cao et al. (2021)) and MEND (Mitchell et al. (2022a)) aim to use hypernetworks to identify the parameters that need to be modified. During prediction, they employ specific methods to output the magnitude of modifications required for those parameters. SERAC (Mitchell et al. (2022b)) achieves knowledge modification by constructing an external memory cache and utilizing a scope classifier to modify the knowledge. (Zheng et al. (2023)) proposes to leverage In-Context Learning (Brown et al. (2020)) to put new knowledge in the prompts to empower models to exploit them. The above methods are for text-only LLMs. Utilizing multimodal data to perform knowledge editing on an MLLM is more in line with real scenarios. The aforementioned methods are all applied to single-modal text-based large models using single-modal data. However, performing knowledge editing on multimodal large language models using multimodal data is more aligned with real-world scenarios. MMEdit (Cheng et al. (2023)) and MIKE (Li et al. (2024)) propose two new multimodal knowledge editing

datasets. However, they do not consider the multi-granularity nature of knowledge in the dataset. Furthermore, their research merely transfers the aforementioned editing methods from LLMs to a specific component in MLLMs. Although they achieved promising performance, we have discovered that simultaneously editing three components can enhance the model's generalization on multimodal data.

## 4.2 MULTIMODAL LARGE LANGUAGE MODEL

Large language models (LLMs) (Brown et al. (2020); Ouyang et al. (2022); Touvron et al. (2023); Zhang et al. (2022)) have demonstrated strong performance on knowledge-intensive tasks (Voorhees & Tice (2000); Talmor et al. (2019); See et al. (2017)). As a result, there have been efforts to train multimodal interfaces in large-scale image caption data for large language models (LLMs) (Alayrac et al. (2022); Li et al. (2023); Zhu et al. (2023); Liu et al. (2023)), enabling them to handle different modalities simultaneously. These models are also known as multimodal large language models (MLLMs) and have shown promising results on knowledge-intensive tasks involving multiple modalities, such as visual question answering (Marino et al. (2019); Antol et al. (2015)) and multimodal dialogue (Wang et al. (2021); Zheng et al. (2022)). These models typically consist of three components: a modality encoder for encoding data from modalities other than text (such as visual encoders), a multimodal interface for transforming representations from other modalities into the space of the LLM, and an LLM, which handles inputs from different modalities along with text inputs to process multimodal tasks. Our method edits knowledge of all components in the MLLM collaboratively and we also analyze the distribution of different knowledge across these components.

## 5 CONCLUSION

In this paper, we introduce a multimodal model editing dataset **M2Edit** for the problem that existing datasets lack multi-granular knowledge, with three types of knowledge: entity, relation, and action. In addition, To address the issue of insufficient generalization of existing methods on multimodal data, we propose the Multimodal Location-based Method (**MLE**). Experiments demonstrated the effectiveness of our method. Additionally, the experiments revealed inconsistencies in the storage regions of different types of knowledge within the MLLM.

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

## A  CASUAL MEDIATION ANALYSIS

Causal mediation analysis aims to identify the causal relationship between different intermediate states in models and the final output of the answer. To trace the important state of the model always needs to take three runs: a clean run that the model can answer the question correctly with normal input, a corrupted run that corrupts the input to make the model get corrupted output, a corrupted-with-restoration run that restores a certain state to judge the restoring of the output.

**Clean Run:** For a sample $(x_i, v_i, y_i) \in \mathcal{D}$, a clean run directly obtains the final answer $(\hat{y}_i)$ through the original MLLM ($\Theta$), which is $\mathbb{P}(y_i) = \Theta(x_i, v_i)$. The state representation of each layer in LLM can be $\mathbf{H}_{llm} = \{h_{llm}^{(i,l)} | i \in [1, T_{llm}], l \in [1, L_{llm}]\}$, where $T_{llm}$ denotes the input token length, $L_{llm}$ denotes the layer numbers of LLM. The same formula holds for the state representation in the visual encoder ($\mathbf{H}_{ve}$) and the multimodal interface ($\mathbf{H}_{mi}$).

**Corrupted Run:** In the corrupted run, the corrupted output ($o$) is obtained by adding Gaussian noise to the input image, which can be expressed as $\mathbb{P}_{cor}(y_i) = \Theta(x_i, v_i + \epsilon)$. The state representation of each layer in different components of MLLM change to be $\hat{\mathbf{H}}_c$, $c \in \{llm, ve, mi\}$.

**Corrupted-with-restoration Run:** In the corrupted-with-restoration run, it replaces each state representation in each component of the corrupted run to clean run. In this way, we can get the new prediction of $y_i$ as $\mathbb{P}_{h_c^{(i,l)}}(y_i) = \Theta_{clean\ h_c^{(i,l)}}(x_i, v_i + \epsilon)$, $c \in \{llm, ve, mi\}$. The indirect effect (IE) of each state representation $h_c^{(i,l)}$ can be: $IE = \mathbb{P}_{h_c^{(i,l)}}(y_i) - \mathbb{P}^{cor}(y_i)$. Averaging over a sample of statements can obtain the average indirect effect (AIE).

## B  DATASET ANNOTATION PROCESS

As illustrated in Figure 7, the annotation process for our method can be broadly divided into three stages: **Data Filtering**, **Diverse Generation**, and **Quality Control**.

**Data Filtering**. Raw data is filtered based on specific rules, which are generally defined as follows: For entity data, each entity must be associated with more than five images, and for relation data, the head entity must have more than three associated images. The image resolution must exceed $64 \times 64$ pixels. For entity data, entity names must consist of a single word. Similarly, for relation data, tail

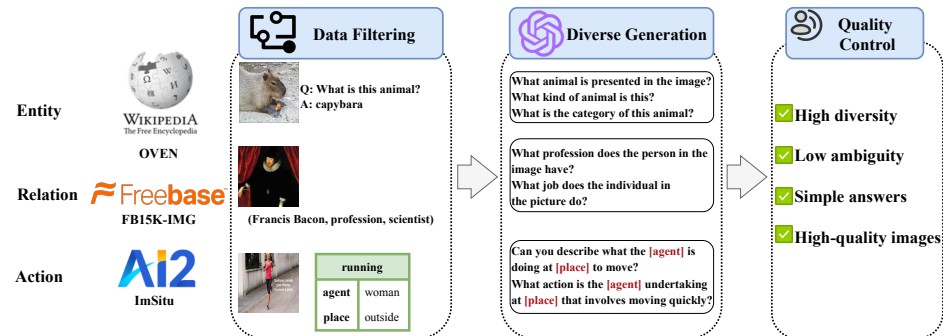

Figure 7: Data Annotation Process Flowchart. First, raw samples of Entities, Relations, and Actions are filtered from Oven, FB15K-IMG, and ImSitu based on predefined rules. Next, the raw data is transformed into QA-form datasets using ChatGPT, incorporating diverse variations. Finally, high-quality data is manually curated to construct the **M2Edit** dataset.

entity names must also be single words. The number of samples within each subclass (defined by entity types, relation terms, or action terms) must exceed 100 samples.

**Diverse Generation**. ChatGPT is employed to generate questions based on relation terms and action frameworks, as illustrated in Figure 7. Additionally, it is instructed to produce synonymous variations of these questions.

**Quality Control**. Finally, the generated questions and their associated samples are manually screened based on the following criteria:

- **High diversity**: The generated questions must exhibit significant variability and avoid mere truncations or expansions.
- **Low ambiguity**: Relation terms and action terms must be distinct, and the generated answers should be as unique as possible.
- **Simple answers**: Answers should be concise (preferably a single word) and should avoid abstract vocabulary.
- **High-quality images**: Images should be diverse, and the content should not contain unclear text or other low-quality elements.

By following this process, we constructed our dataset **M2Edit**.

