# OpenReview forum: "M2Edit: Locate and Edit Multi-Granularity Knowledge in Multimodal Large Language Model"
_ICLR.cc/2025/Conference — Submitted to ICLR 2025_

### Official Review · Reviewer_M9Ma · 2024-10-21

**Soundness:** 2
**Presentation:** 1
**Contribution:** 2
**Rating:** 3
**Confidence:** 5

**Summary:**

The paper explores advanced techniques in knowledge editing for multimodal large language models (MLLMs).
The authors introduce M2Edit, a dataset to enhance multimodal knowledge editing by incorporating multi-granularity knowledge types—entities, relations, and actions.
They highlight the limitations of existing methods and datasets in addressing multi-granularity and multimodal challenges.
The paper proposes a method, MLE (Multimodal Location-based Editing), which improves knowledge editing by identifying and modifying key knowledge layers across the various components of MLLMs.
Experiments show that the method improves visual generality performance and achieves superior results on multi-granularity knowledge compared to existing benchmarks.

**Strengths:**

- The reviewer finds the authors' motivation for considering multi-granularity in knowledge editing to be interesting, which aligns with intuitive understanding of the subject.
- The dataset contributed by the authors (if executed flawlessly) could potentially be a significant asset to the field of multimodal knowledge editing.
- The method introduced by the authors outperforms the baseline.

**Weaknesses:**

The paper's primary issue appears to be the unclear expression and presentation of content, with so many details lost, making it difficult for the reviewer to understand the whole story fully. Furthermore, significant potential issues have been identified within the dataset section.


## Major Concerns:
- The reviewer questions the setting of multimodal knowledge editing as posited by the authors, perceiving that it remains limited to textual LLM thinking. Notably, the M2Edit only edits the textual part of image-text pairs, which implies no equivalent editing for visual knowledge, thereby not affecting the image component. If the entire multimodal knowledge editing topic is defined in this manner, the reviewer questions the scientific validity of this definition and suggests that a fundamental improvement is necessary. The reviewer suggests that the authors further clarify this point in their discussion.
- While the paper mentions relational type knowledge in a triplet format, the examples shown in Figures 1 and 2 do not represent triplets but rather entity-level knowledge. The manifestation of relation-level knowledge editing remains unclear. The reviewer recommends that the authors revise and clarify this point clearly in the text.
- The dataset claims the importance of three levels of knowledge but does not integrate these levels within a single scope; different levels of annotations cannot coexist within the same instance, which likely limits the dataset's utility. Therefore, the reviewer hopes that the authors can further explain and clarify this matter.
- The data annotation process is not clearly articulated, raising concerns about the control over data quality, especially as it relies entirely on an automated process via ChatGPT, which is prone to introducing noise. Please provide a detailed description of this step in the manuscript.
- Figure 2 is really challenging to understand; it is unclear what the multiple lines of text within circles represent. Please provide further details.
- Similarly, Figure 3 is also difficult to decipher; the meanings of various arrows and shapes within the figure are not explained, and the significance of the different rectangles in the bottom-left box and what r, s, t represent are not clarified. Please provide additional information.
- In the methods section, the authors claim that to address the limitations of existing knowledge editing methods—which cannot handle multi-granularity knowledge and lack generalization on multimodal data—they propose a method called MLE (Multimodal Location-based Editing). However, the reviewer does not understand the causal relationship between the existing methods' inability to handle multi-granularity knowledge and the proposed "Locate Then Edit" approach. Is it necessary for multi-granularity knowledge editing to be implemented specifically through a "Locate Then Edit" method?
- The methods were only validated using older MLLMs like BLIP2-OPT and MiniGPT4, which may not represent the most advanced MLLMs, thus not sufficiently proving the effectiveness and generality of the proposed multimodal knowledge editing methods. The reviewer suggests adding more MLLMs for experimental comparison.
- The experimental analysis conducted by the authors lacks sufficient depth and breadth. The reviewer strongly recommends enhancing the content of the experimental analysis.
- The absence of any anonymous links for accessing model code and data examples impedes the reviewer's ability to further investigate and address the issues raised, casting doubts on the reproducibility of the research. Will the authors consider open-sourcing the code and resources?



## Typos & Expression

Overall, the writing and expression in the paper are overly casual and lack the refinement expected in scholarly communication.

- There is a grammatical error on page three, line 112.
- All images in the paper are non-vectorial.
- The citation format throughout the paper does not adhere to standard academic norms.
- There are numerous detail-oriented issues, such as inconsistent punctuation in equations—some equations end with a comma or period while others do not, creating a disorganized appearance.


Overall, the reviewer is open-minded. If the authors can actively and effectively address these concerns, the reviewer would consider raising the rating.

**Questions:**

The paper contains quite many aspects that are not clearly explained, making it challenging for the reviewer to understand. Below are some questions that need addressing:


- The term "in-scope" mentioned in Figure 2 and its caption is ambiguous; does it refer to "in-domain"?
- The caption in Figure 2 states, "After editing the MLLMs, the in-scope samples need to be generalizable, and the out-of-scope samples should not be unchanged", but this statement is confusing and lacks clarity.
- The head entity "Arcadia" mentioned on page four, line 177, is not visible in the middle part of Figure 2, making the reviewer confused about its inclusion and relevance.
- Beyond BLIP2-OPT and MiniGPT4, how does the proposed method perform on other state-of-the-art MLLMs?

---

> ### Author Response · Authors · 2024-11-21
> **Response to Reviewer #M9Ma (1/2)**
>
> Thank you for providing valuable feedback on our paper. Below, we have addressed your comments in detail and hope our responses sufficiently clarify the points you raised. Should you have any further questions or concerns, please do not hesitate to let us know. (please note that we break our response into two parts due to space constraints)
>
> ## Q1: ```Exploring broader multimodal knowledge editing tasks.```
> > **Knowledge itself is modality-agnostic**[6], with text and images serving as different representations of the same information. Currently, the construction of multimodal large language models (MLLMs)[5] predominantly centers on large language models (LLMs), with additional encoders (e.g., visual encoders) integrated through representation alignment. As a result, most MLLMs produce textual outputs, and existing definitions of knowledge editing tasks in these models adhere to this paradigm. In contrast, image editing[7] constitutes a separate research direction, focusing on editing visual concepts in generated images, which targets different types of models. Expanding to broader knowledge editing tasks, as you suggest, would require further development of specific application scenarios and datasets to advance this line of research.
> ------
>
> ## Q2: ```Definitions of entity-level, relation-level, and event-level knowledge```.
>
> > In constructing knowledge graphs, knowledge is typically categorized into three types[1] :
> >
> > * **Entity knowledge**: Refers to information about specific objects in the real or conceptual world. These objects can be tangible, like "Apple Inc." or "Mount Huangshan," or abstract, like "love" or "economics." Entities can be represented in textual or visual form. For instance, as illustrated in the left panel of Figure 2, the concept of "capybara" can be represented either as the word *Capybara* or as its corresponding image. This panel essentially represents the triplet *(Image of capybara, is a, capybara)*.
> > * **Relational knowledge**: Represents the semantic relationships between entities, such as interactions and associations. For example, the middle panel of Figure 2 depicts a locational relationship, representing the triplet *(Image of Arcadia, State, California)*.
> > * **Event knowledge**: Concerns activities involving one or more participants (agents) in a specific spatiotemporal context, centered around a particular theme. For instance, the right panel of Figure 2 demonstrates the concept of the action "Running," representing its structural relationships with the associated "agent" and "place".
> ------
>
> ## Q3: ```Addressing the integration of all three types of knowledge.```
>
> > Figure 1 illustrates a real-world scenario encompassing all three types of knowledge. In our experiments, we observed that different types of knowledge are stored in different regions of the model. While unified editing approaches achieve some success, they lack sufficient performance in multimodal generalization and overall efficacy. Therefore, we believe targeted editing is essential. Developing datasets that closely resemble real-world scenarios remains challenging. Ensuring that a single query encompasses all three types of knowledge, that **none of these types are pre-existing in the MLLM**, and that test cases consistently evaluate the model's edited capabilities is a complex task. Nevertheless, our proposed task addresses gaps in prior work by exploring knowledge across multiple granularities.
> ------
>
> ## Q4: ```Clarifications on the data annotation process and ChatGPT's involvement.```
>
> > As stated in Section 2.2, the M2Edit dataset primarily builds on existing datasets such as Oven[2], FB15k-237-IMG[3], and ImSitu[4], which have been validated for data quality in prior studies. To suit our needs, we constructed the dataset through automated methods combined with manual filtering. To enhance question diversity, we used ChatGPT to generate questions, but all generated questions were manually reviewed and filtered to ensure data quality. In future versions of the paper, we will include a diagram of the annotation process in the appendix for greater clarity.
> ------
>
> **Referrence**
>
> [1]  Entity, Relation, and Event Extraction with Contextualized Span Representations. EMNLP 2019.
>
> [2] Open-domain visual entity recognition: Towards recognizing millions of wikipedia entities. ICCV 2023.
>
> [3] MMKG: multi-modal knowledge graphs. ESWC 2019.
>
> [4] Situation recognition: Visual semantic role labeling for image understanding. CVPR 2016.
>
> [5] A Survey on Multimodal Large Language Models for Autonomous Driving. WACVW 2024.
>
> [6] Multi-Modal Knowledge Graph Construction and Application: A Survey. IEEE Trans. Knowl. Data Eng.
>
> [7] Diffusion Model-Based Image Editing: A Survey. Arxiv 2024.

---

> ### Author Response · Authors · 2024-11-21
> **Response to Reviewer #M9Ma (2/2)**
>
> ## Q5: ``Difficulties in understanding Figure 2.``
>
> > We apologize for any confusion regarding Figure 2. To address this, we have provided more detailed explanations in **General Response Q1**, which we invite you to review for further clarification.
>
> ------
>
> ## Q6: ```Explanation of Figure 3 and the meaning of s, r, and t.```
>
> > In Figure 3 (lines 216-235), the top-left section illustrates the architecture of an MLLM[8,9,10,11], which consists of three main components: the **Visual Encoder**, **Multimodal Interface**, and **Large Language Model**. The bottom-left parallelogram sequences represent the layers of these components. Variables $s$, $r$, and $t$ denote the **Key Knowledge Layers**, as referenced in Equation 6 (lines 247-249), which are the layers most relevant to the outputs for the corresponding knowledge types. These layers are the focus of our editing process, as illustrated in the bottom-right editing diagram. The top-right section outlines four evaluation dimensions for MLLM knowledge editing, as defined in Equations (1)-(4) in our paper.
>
> ------
>
> ## Q7: ``The absence of causality handling in multi-granularity knowledge editing and the necessity of a locate-then-edit approach for multimodal large language models (MLLMs).``
>
> > As discussed in the Introduction of our paper, existing knowledge editing methods[12,13] do not simultaneously address editing across different components of multimodal large language models. This limitation reduces the models' generalization ability after editing (see Table 2, lines 324–347). Our approach, in contrast, edits three distinct components simultaneously. Furthermore, during the localization process, we identified that different types of knowledge are stored in different regions of the model (see Figure 4, lines 378–389). For this reason, we believe that localization is essential for effective knowledge editing, which experimental results have also verified.
>
> ----
>
> ## Q8: ``Validation limited to older MLLMs (e.g., BLIP2-OPT and MiniGPT4), which may not represent state-of-the-art models.``
>
> > Thank you for this valuable suggestion. In response, we conducted additional experiments with LLaVa-7B [11]. The results are presented below:
>
> > $$\begin{array}{lccc}
>  \hline
> \textbf{Method} &\textbf{Entity}                  & \textbf{Relation}                       &                    \textbf{Action}                          \\\\
> \hline
> \textbf{FT} &33.5& 27.0& 30.2\\\\
> \textbf{MEND} & 82.3 &70.6&77.8 \\\\
> \textbf{ROME} &71.2& 52.5	& 78.3  \\\\
> \hline
> \textbf{MLE} & \textbf{86.4}	& \textbf{75.9} &	\textbf{86.0}
> \end{array}$$
>
> > This table summarizes the comprehensive editing results of different methods applied to LLaVa-7B [11] on our dataset. It demonstrates that our method continues to exhibit superior generalization performance.
>
> ----
>
> ## Q9: ``Suggestions to enhance experimental analysis.``
>
> > Thank you for your insightful suggestion. In our current paper, we have already shown that our method outperforms baseline models in both comprehensive performance and multimodal generalization across various types of knowledge. Additionally, we have analyzed the distribution of different knowledge types within the model, which supports the superiority of our approach. To further strengthen our work, we have added the results of batch editing experiments (see **General Response Q1**) and plan to include a case study analysis in the appendix of the updated version.
>
> ----
>
> ## Q10: ``Missing anonymous links to access model code and data examples.``
>
> > We appreciate this suggestion. During the initial submission, we provided data examples. We have now updated the supplementary material to include the MLE code in the attachments.
>
> ----
>
> ## Q11: ```The meaning of in-scope editing.```
>
> > In-scope editing refers to the scope of content that should be modified during a single editing operation. This concept was introduced in [reference]. For in-scope content, all relevant elements must be modified, while out-of-scope content should remain unaffected. Detailed examples of in-scope and out-of-scope editing are provided in **General Response Q1**.
>
> ----
>
> ## Q12: ```Grammatical and punctuation errors.```
>
> > Thank you for pointing out the grammatical and punctuation issues. We sincerely appreciate your careful review and constructive feedback. We will address these issues thoroughly to meet ICLR standards.
>
> ----
>
> **Referrence**
>
> [8]  InstructBLIP: Towards General-purpose Vision-Language Models with Instruction Tuning. NeurIPS 2023.
>
> [9] Flamingo: a Visual Language Model for Few-Shot Learning. NeurIPS 2023.
>
> [10] MiniGPT-4: Enhancing Vision-Language Understanding with Advanced Large Language Models. ICLR 2024.
>
> [11] Improved Baselines with Visual Instruction Tuning. CVPR 2024.
>
> [12] Can We Edit Multimodal Large Language Models? EMNLP 2023.
>
> [13] MIKE: A New Benchmark for Fine-grained Multimodal Entity Knowledge Editing. ACL 2024.

---

> ### Author Response · Authors · 2024-12-03
> **Response to Reviewer #M9Ma (2/2)**
>
> ## Q4: ```Provide more details about the annotation process.```
>
> > Details regarding the annotation process were already present in the original manuscript. To improve clarity, we have added further explanations in the appendix (**Lines 748–791**) and introduced a new **Figure 7** (**Line 756**). We hope these additions address your concerns.
>
> ---
>
> ## Q5: ```Update image content.```
>
> > Thank you for pointing this out. We have updated all images in the manuscript to clearer vector graphics, adhering to your suggestion.
>
> ---
>
> ## Q6: ```The necessity of locate-then-edit for multi-granularity knowledge editing.```
>
> > We would like to clarify the rationale behind our proposed approach. While our locate-then-edit strategy is a means to achieve multi-granularity knowledge editing, it is not the **only** possible method. The key contributions of our work, which distinguish it from prior studies, are as follows:
>
> > 1. **Editing across multiple components:** Unlike prior methods that edit a single component of an MLLM, our approach edits the visual encoder, multimodal interface, and large language model collaboratively, resulting in superior generalization capabilities (**See Figure 4**).
>
> > 2. **Feasibility through localization:** Our approach is grounded in parameter adjustment for specific layers (following the methodology of [3]). Without precise localization, global fine-tuning would result in excessive interference with unrelated knowledge.
>
> > 3. **Knowledge distribution analysis:** Our analysis identifies the uneven distribution of knowledge across components, substantiating the necessity of targeted edits (**See Figure 5** and **Lines 378–403**).
>
> ---
>
> ## Q7 & Q8: ```Experimental limitations.```
>
> > Despite limited resources, we have supplemented our experiments with:
>
> > 1. **Batch editing results.**  (paper line 432-446 and line 459-466)
> > 2. **Performance on different sizes of LLaVa models.**
>
> > Previous studies ([1,2,3]), regardless of modality, typically evaluated medium-scale models. To ensure rigor, we extended our evaluations to various LLaVa configurations (Edit Score), with results summarized below:
> > $$\begin{array}{llccc}
> \hline
> \textbf{Model} & \textbf{Method} &\textbf{Entity}                  & \textbf{Relation}                       &                    \textbf{Action}                          \\\\
> \hline
> \textbf{LLaVa 1.5 7B } & \text{FT} &33.5	 & 27.0	& 30.2\\\\
> & \text{MEND} & 82.3 &	70.6	& 77.8 \\\\
> & \text{ROME} &71.2	& 52.5	& 78.3  \\\\
> & \textbf{MLE} & \textbf{86.4}	& \textbf{75.9} &	\textbf{86.0} \\\\
> \hline
> \textbf{LLaVa 1.6 7B } & \text{FT} &18.7	 & 15.2	& 24.3\\\\
> & \text{MEND} & 79.5 &	63.3	& 81.0 \\\\
> & \text{ROME} &75.1	& 61.8	& 76.4  \\\\
> & \textbf{MLE} & \textbf{81.2}	& \textbf{70.2} &	\textbf{83.6} \\\\
> \hline
> \textbf{LLaVa 1.6 13B } & \text{FT} &42.5	 & 36.2	& 43.6\\\\
> & \text{MEND} & 85.2 &	78.9	& 84.7 \\\\
> & \text{ROME} &81.5	& 72.4	& 84.6  \\\\
> & \textbf{MLE} & \textbf{89.2}	& \textbf{79.9} &	\textbf{86.8} \\\\
> \hline
> \textbf{LLaVa 1.6 34B } & \text{FT} &53.2	 & 37.7	& 44.5\\\\
> & \text{MEND} & 87.0 &	79.2	& 85.3 \\\\
> & \text{ROME} &82.2	& 74.4	& 85.0  \\\\
> & \textbf{MLE} & \textbf{88.4}	& \textbf{79.5} &	\textbf{85.9} \\\\
> \hline
> \end{array}$$
> > The **Edit Score** was computed as:
> > $$
> \textbf{Edit Score} = \frac{4}{\frac{1}{\mathbf{O}^{rel}} + \frac{1}{\mathbf{O}^{gen} _ v} + \frac{1}{\mathbf{O}^{gen} _ t} + \frac{1}{\mathbf{O}^{loc}}}.
> $$
> > These results demonstrate the robustness of our method across diverse multimodal models.
>
> ---
>
> In conclusion, we believe our current experiments and settings adequately demonstrate the effectiveness of our method. We will strive to incorporate further analyses and insights based on your feedback in future revisions.
>
> Thank you for your invaluable comments!
>
>
>
> ----
>
> **References**
>
> [1] Can we edit multimodal large language models? EMNLP 2023
>
> [2] MIKE: A New Benchmark for Fine-grained Multimodal Entity Knowledge Editing. ACL 2024.
>
> [3] MC-MKE: A Fine-Grained Multimodal Knowledge Editing Benchmark Emphasizing Modality Consistency. Arxiv 2024.

---

### Official Review · Reviewer_ddNb · 2024-10-28

**Soundness:** 3
**Presentation:** 3
**Contribution:** 3
**Rating:** 6
**Confidence:** 2

**Summary:**

The paper introduces M2Edit, a dataset with entity, relation, and action knowledge types, designed for multimodal large language models (MLLMs). It highlights the challenge of knowledge editing across different granularities within MLLMs and proposes the MLE (Multimodal Location-based Editing) method to tackle this. MLE sequentially identifies and edits key knowledge layers within MLLMs, enhancing generality and effectiveness in multimodal contexts. The model demonstrates improved accuracy and generalization over previous methods.

**Strengths:**

- Innovative multi-granularity approach in knowledge editing for MLLMs, addressing a gap in existing datasets.
- The MLE method shows significant performance improvements, particularly in visual generality and model adaptability.
- Offers detailed methodology for locating and editing specific knowledge layers within MLLMs, aiding model interpretability.

**Weaknesses:**

The proposed method is evaluated on a limited range of multimodal models, which restricts the generalizability of the findings across other MLLMs with different architectures or training objectives. Specifically, the recent VL models like QwenVL2, Llava should be evaluated.

**Questions:**

No

---

> ### Author Response · Authors · 2024-11-14
> **Response to Reviewer #ddNb**
>
> We appreciate very much your constructive comments on our paper. Please kindly find our response to your comments below. We hope that our response satisfactorily addresses the issues you raised. Please feel free to let us know if you have any additional concerns or questions.
>
> ## Q1:  ``` The proposed method is evaluated on a limited range of multimodal models. ```
>
> > - In line with the methodologies employed in previous studies [1], which focused on Mini-GPT4 and BLIP-2, we have maintained their experimental framework to uphold the integrity of our research. To substantiate the efficacy of our approach, we have extended our investigation with further experiments:
>
> > - Evaluation of the Edit Score for LLaVA-1.5 7B [2,3]:
>
>
> >$$ \begin{array}{lccc}
> \hline
> \textbf{Method} &\textbf{Entity}                  & \textbf{Relation}                       &                    \textbf{Action}                          \\\\
> \hline
> \textbf{FT} &33.5	 & 27.0	& 30.2\\\\
> \textbf{MEND} & 82.3 &	70.6	& 77.8 \\\\
> \textbf{ROME} &71.2	& 52.5	& 78.3  \\\\
> \hline
> \textbf{MLE} & \textbf{86.4}	& \textbf{75.9} &	\textbf{86.0}
> \end{array} $$
>
> >The results clearly demonstrate the superior performance of our method when applied to LLaVA-1.5 [2,3].
> -------
>
> **References**
>
> [1] Exploring Edits in Multimodal Large Language Models. EMNLP 2023.
>
> [2] Enhancing Visual Instruction Tuning. NeurIPS 2023.
>
> [3]Improved Baselines with Visual Instruction Tuning. CVPR 2024.

---

> > ### Comment · Reviewer_ddNb · 2024-11-23
> >
> > Thanks for the response. Although the provided table is not clearly visible, it helps authors' claim.

---

> > > ### Author Response · Authors · 2024-12-03
> > > **Response to Reviewer #ddNb**
> > >
> > > We were previously unaware that the paper could be modified during the review process. Based on your valuable suggestions, we have made partial revisions to the manuscript and highlighted the changes in blue for your convenience. We sincerely hope you find these updates helpful. Thank you very much for your understanding and support!

---

### Official Review · Reviewer_yLfA · 2024-11-04

**Soundness:** 3
**Presentation:** 3
**Contribution:** 3
**Rating:** 6
**Confidence:** 4

**Summary:**

The paper works on multimodal knowledge editing.
It introduces a new dataset for this task, M2Edit (Multi-Granularity Multimodal knowledge Editing dataset). This dataset incorporates multi-granularity knowledge (relation, entity, and action) to address the limitations of existing multimodal knowledge editing datasets.
Moreover, the paper proposes a multimodal knowledge editing method, MLE (Multimodal Location-based Editing).
It identifies key knowledge layers within different components of MLLMs and collaboratively edits them to improve the model's performance on multimodal data. The method demonstrates significant improvements in visual generality performance and performs well in terms of different knowledge granularities.

**Strengths:**

1. The paper is overall clear and well-organized.
2. The paper proposes a useful dataset M2Edit including multi-granularity knowledge. This addresses the limitations of previous multimodal knowledge editing datasets.
3. The proposed method MLE can edit multi-granularity knowledge within MLLMs.
4. The report experiments verify the performance of MLE.

**Weaknesses:**

1. The paper ignores the discussion on the complexity of the MLE method.
2. The paper currently has limited analysis of error cases. Adding this could inspire further research work.
3. The uploaded material doesn't include the code, only the used dataset.
4. The paper doesn't mention how many samples the method edits at once, so it seems the paper does not report the results of batch editing.
5. The introduced dataset M2Edit seems to include counterfactual knowledge. How does the MLE perform with real-world knowledge as in [1,2]?

[1] MQuAKE: Assessing Knowledge Editing in Language Models via Multi-Hop Questions
[2] Updating Language Models with Unstructured Facts: Towards Practical Knowledge Editing

**Questions:**

1. Line 483, To address --> to address
2. It is hard to recognize the sentences with the striped background in Figure 2.
3. Can you provide some analysis of error cases?
4. The supplementary material only contains the M2Edit dataset. What about the code of MLE?
5. The paper should explain how the various metrics are computed.
6. How many samples do you edit at once? What is the performance of MLE when editing with several samples?

---

> ### Author Response · Authors · 2024-11-22
> **Response to Reviewer #yLfA (1/2)**
>
> Thank you for your thoughtful feedback. Below, we provide detailed responses to each of your questions and concerns. (please note that we break our response into two parts due to space constraints)
>
> ------
> ## Q1: ```Lack of discussion on the complexity of the method.```
>
> > In prior work on knowledge editing, the complexity of such methods has seldom been emphasized. This is likely because in real-world scenarios, frequent and large-scale editing is uncommon. Additionally, editing model parameters through knowledge editing involves significantly fewer parameters than fine-tuning, making it much more efficient—our method is over 5x faster than fine-tuning. Specifically, editing 500 samples from our dataset takes only 1/50th the time required for fine-tuning.
>
> > Although our method involves clustering and similarity calculations, clustering is performed only once beforehand. During testing, the time spent on similarity calculations is negligible compared to the time required to adjust model parameters. Below is a table showing the time required for editing 500 M2Edit samples on BLIP2-OPT 7B [1] using NVIDIA GeForce RTX 3090 GPUs:
> > $$
> \begin{array}{lc}\hline\textbf{Method} &\textbf{Time(s)}\\\\ \hline \textbf{FT} & 1617.5 \\\\\textbf{ROME} & 13.1 \\\\\hline\textbf{MLE} & 29.2                            \end{array}
> $$
> ------
>
> ## Q2: ```Lack of discussion on failure cases.```
>
> > Due to the limitations of displaying image content on OpenReview, we will include updated case studies in the next version of the paper. In general, traditional methods such as MEND [2] and ROME [3] struggle to generalize well to changes in images after editing a specific component.
>
> > For example, when editing entity-level knowledge about "Taco," traditional methods can generalize text queries from "What is the name of this dish?" to "What is this food called?" (T-G) and still output "Taco." However, they fail when dealing with synonymous but visually distinct images of "Taco" (V-G). In contrast, our method can correctly answer in such scenarios, demonstrating superior visual generality.
>
> ------
>
> ## Q3: ```Only dataset was uploaded; no code provided.```
>
> > Thank you for pointing this out. While we submitted dataset samples during the initial submission, we have now updated the supplementary material to include the MLE code in the attachments.
>
> ------
>
> ## Q4: ```Lack of batch editing results.```
>
> > We appreciate your suggestion. Batch editing results and analyses have been added and can be found in **General Response Q1**.
>
> ------
>
> ## Q5: ```Lack of results on real-world datasets.```
>
> > Thank you for the suggestion. While the two valuable references you provided are focused on text-only modalities, our method specifically targets multimodal large language models. This difference makes validation on these datasets challenging. The key challenge in multimodal knowledge editing lies in the lack of real-world, high-quality datasets for evaluation [4,5]. Should such datasets become available, we would be delighted to validate our method on them.
>
> ------
>
> **References**
>
> [1] BLIP-2: Bootstrapping Language-Image Pre-training with Frozen Image Encoders and Large Language Models. ICML 2023.
>
> [2] The Tenth International Conference on Learning Representations. ICLR 2022.
>
> [3] Locating and Editing Factual Associations in GPT. NeurIPS 2022.
>
> [4] Can we edit multimodal large language models? EMNLP 2023.
>
> [5] MIKE: A New Benchmark for Fine-grained Multimodal Entity Knowledge Editing. ACL 2024.

---

> ### Author Response · Authors · 2024-11-22
> **Response to Reviewer #yLfA (2/2)**
>
> ## Q6: ```Issues with grammar and clarity in figure presentation.```
>
> > We apologize for the lack of clarity in Figure 2. We have provided a detailed explanation of its content in **General Response Q2** and encourage you to review it.
>
> ------
>
> ## Q7: ```Metric calculation issues.```
>
> > The calculation methods for **Reliability (R)**, **Visual Generality (V-G)**, **Text Generality (T-G)**, and **Locality (L)** are detailed in Equations (1)–(4) of our paper. Below, we briefly summarize their computation:
>
> > - **Reliability (R):** Measures accuracy on edited samples $(x_i, v_i, y_i)$.
>
> >  $$ \mathbf{O}^{rel}(\hat{\Theta}) = \mathbb{E}_{(x_i,v_i,y_i) \in \mathcal{D}} [\mathbf{I}( \hat{\Theta}(x_i,v_i) = y_i) ]. $$
>
> > - **Text Generality (T-G):** Tests the edited sample with synonymous text $x_j$ paired with the original image $v_i$.
>
> >  $\mathbf{O}^{gen}(\hat{\Theta}) = \mathbb{E}_{(x_i,v_i,y_i) \in \mathcal{D}} [\mathbf{I}( \hat{\Theta}(x_j,v_i) = y_i) ], s.t. ~ xj\sim xi.$
>
> > - **Visual Generality (V-G):** Tests the edited sample with synonymous images $v_j$ paired with the original text $x_i$.
>
> >  $ \mathbf{O}^{gen}(\hat{\Theta}) = \mathbb{E}_{(x_i,v_i,y_i) \in \mathcal{D}} [\mathbf{I}( \hat{\Theta}(x_i,v_j) = y_i) ], s.t. ~ vj \sim vi.$
>
> > - **Locality (L):** Measures the impact of editing on unrelated samples $(x_k, v_k)$.
>
> >  $$ \mathbf{O}^{loc}(\hat{\Theta}) = \mathbb{E}_{(x_k,v_k) \in \mathcal{D}} [\mathbf{I}( \hat{\Theta}(x_k,v_k) = \Theta(x_k,v_k)) ],
>    s.t.~(x_k,v_k) \perp (x_i,v_i). $$
>
> We hope our responses address your concerns adequately. Please feel free to share any additional feedback or questions you may have. Thank you again for your thoughtful and constructive comments!

---

> > ### Comment · Reviewer_yLfA · 2024-11-26
> > **Response to authors**
> >
> > Thank the authors for the response. I'd like to maintain my positive score.

---

> > > ### Author Response · Authors · 2024-12-03
> > > **Response to Reviewer #yLfA**
> > >
> > > We were previously unaware that the paper could be modified during the review process. Based on your valuable suggestions, we have made partial revisions to the manuscript and highlighted the changes in **blue** for your convenience. We sincerely hope you find these updates helpful. Thank you very much for your understanding and support!

---

### Author Response · Authors · 2024-11-19
**General Response**

## Q2: ``Regarding the clarity of Figure 2.``

We sincerely apologize for the issues with Figure 2 (line 162-173). During submission, embedding vector graphics caused compilation errors, and design flaws made the figure difficult to read. We deeply regret this and would like to provide a detailed explanation of Figure 2 for the reviewers’ reference. We greatly appreciate your understanding.

> The data setup in our paper largely follows prior work \[1,2] on knowledge editing in multimodal large language models. In this context:
>
> * **Edit target** refers to the knowledge that needs to be edited. To ensure this knowledge does not already exist in the multimodal language model and to rigorously test the editing capability, all edit targets in M2Edit involve counterfactual knowledge. For example, in the case of entity editing (left panel of Figure 2), the original knowledge about "capybara" is edited so the model’s output should instead be "koala."
>
> - **In-scope** refers to all content that should be altered by a single edit.
> - **Out-of-scope** refers to content that should remain unaffected by the edit.



> ### Entity Editing (Left Panel)
>
> The four images in this panel represent the concept of "capybara." The questions—“What animal is presented in the image?”, “What is this animal?”, “What kind of animal is this?”, and “What is the category of this animal?”—probe the entity's identity. The editing target aims for the model to consistently output "koala" regardless of which image of "capybara" is used or which phrasing is applied in the question. At the same time, for unrelated entity knowledge, such as identifying a “library” from its image and answering “What is this place?”, the model should still correctly respond with “library” after the edit.



> ### Relation Editing (Middle Panel)
>
> The four images depict the concept of "Arcadia." The questions—“In which state is the location depicted in the image situated?” and “Can you identify the state where the place in the picture is located?”—probe the location relation (state). The editing target ensures that, after the edit, the model consistently outputs "Louisiana" for any image of "Arcadia" and any phrasing of the location-relation question. Simultaneously, for unrelated relation knowledge, such as identifying the occupation of "Peter Morgan" from his image and answering “What job does this individual in the picture do?”, the model should still output “actor.”



> ### Action/Event Editing (Right Panel)
>
> The four images illustrate the concept of "running." The questions—“Can you describe what the [agent] is doing at [place] to move?” and “What action is the [agent] undertaking at [place] that involves moving quickly?”—probe the action being performed. Here, the semantic slots (e.g., "[agent]" and "[place]") are filled based on annotations for images  from the ImSitu dataset [3], such as "[agent]" being "woman" and "[place]" being "outside." For all in-scope questions and images, the edited model should output "sitting" instead of "running." However, for out-of-scope action queries, such as identifying the action “shaving” from an image, the model should still output “shaving” post-edit.

We hope this explanation clarifies Figure 2. Thank you again for your patience and understanding.

---

**Reference**

[1] Can we edit multimodal large language models? EMNLP 2023.

[2] MIKE: A New Benchmark for Fine-grained Multimodal Entity Knowledge Editing. ACL 2024.

[3] Situation recognition: Visual semantic role labeling for image understanding. CVPR 2016.

---

### Author Response · Authors · 2024-11-20
**General Response**

## Q1: ```Batch editing results.```

$$\begin{array}{lcccclcccclcccc}
\hline
\textbf{Method} & & &\textbf{Entity}                   &        &                     && & \textbf{Relation}                       &                    &&  & \textbf{Action}                          \\\\
                                 & \textbf{R}     & \textbf{T-G}  & \textbf{V-G}  & \textbf{L}    &  & \textbf{R}    & \textbf{T-G}  & \textbf{V-G}  & \textbf{L}    &  & \textbf{R}    & \textbf{T-G}  & \textbf{V-G}  & \textbf{L}    \\\\                               \hline
\textbf{\emph{For BLIP2-OPT}}   \\\\ \hline
\textbf{FT}                      & \textbf{67.4} &20.2 &15.6 &26.4 & &53.2 &18.7 &8.5 &40.2 & &\textbf{81.3} & 32.6 &8.9 &43.3         \\\\
\textbf{MEND} & 48.1 &44.2 &32.5 &80.4 & &42.0 & \textbf{38.6} &31.8 &\textbf{83.1} & &73.2 &65.3 &35.4 &90.4                       \\\\
\textbf{ROME} & 45.4 &41.8 &26.9 &82.5 & &38.3 &35.3 &35.0 &79.5 & &76.7 &63.2 &41.2 &\textbf{91.2}                   \\\\ \hline
\textbf{MLE}  & 65.9 &\textbf{45.2} &\textbf{46.3} &\textbf{83.1}& &\textbf{47.2} &37.2 &\textbf{43.3} &80.5 & &77.3 &\textbf{66.8} &\textbf{54.8} &\textbf{91.2}                    \\\\ \hline \\\\ \hline
\textbf{\emph{For MiniGPT4}}                                                                                                                                                                                                                                        \\\\ \hline \hline
\textbf{FT}   & 24.2 &5.8 &5.2 &26.3 &&15.0 &4.7&1.4&38.2&&28.9&22.3&5.4&54.3                            \\\\
\textbf{MEND} & 53.7&50.2&34.4&82.4&&46.7&38.4&24.7&88.2&&63.4&55.3&43.2&92.3                            \\\\
\textbf{ROME} & 55.2 &48.6 &32.4 &\textbf{84.0} &&48.2&39.1&27.2&89.2&&72.3&59.4&48.9&93.3                        \\\\ \hline
\textbf{MLE}  & \textbf{61.3} &\textbf{51.9} &\textbf{43.8} &82.6 & &\textbf{51.3}&\textbf{39.5} &\textbf{34.3}&\textbf{90.1}& &\textbf{74.7}&\textbf{61.5}&\textbf{53.7}&\textbf{93.5}                  \\\\
\end{array}$$

> * The table shows batch editing results (500 edits). From the table, it is evident that our method maintains a significantly strong comprehensive performance compared to the baseline model, especially in terms of multimodal generalization (V-G), outperforming the baseline model across all three knowledge settings.

---

### Meta-Review · Area_Chair_7qkD · 2024-12-15

**Metareview:**

This paper introduces M2Edit, a multimodal knowledge editing dataset encompassing multi-granularity knowledge types (entity, relation, and action) and proposes the MLE method for locating and editing knowledge in multimodal large language models (MLLMs). While the paper addresses an important challenge and demonstrates promising results on visual generality and task-specific improvements, it suffers from several critical issues. The evaluation is limited to older MLLMs, and broader applicability to more advanced models remains unverified. Additionally, the methodology lacks sufficient clarity and theoretical rigor, particularly in the "Locate-Then-Edit" framework and its relationship with multi-granularity knowledge editing. The dataset construction also raises concerns about data quality and diversity, as it relies heavily on automated methods without sufficient manual verification. These weaknesses outweigh the paper's contributions, leading to a recommendation to Reject.

**Additional Comments On Reviewer Discussion:**

The reviewers acknowledged the novelty of addressing multi-granularity knowledge editing but raised concerns about dataset quality, limited validation on advanced models, and the unclear justification for the "Locate-Then-Edit" framework. While the authors provided additional experiments and clarifications during the rebuttal, key issues, including the lack of scalability and methodological clarity, remained unresolved. This ultimately led to a consensus to reject the paper.

---

### Decision · Program_Chairs · 2025-01-22

Reject